# STABILITY ANALYSIS OF SGD THROUGH THE NORMALIZED LOSS FUNCTION

## ABSTRACT

We prove new generalization bounds for stochastic gradient descent for both the convex and non-convex cases. Our analysis is based on the stability framework. We analyze stability with respect to the normalized version of the loss function used for training. This leads to investigating a form of angle-wise stability instead of euclidean stability in weights. For neural networks, the measure of distance we consider is invariant to rescaling the weights of each layer. Furthermore, we exploit the notion of on-average stability in order to obtain a data-dependent quantity in the bound. This data-dependent quantity is seen to be more favorable when training with larger learning rates in our numerical experiments. This might help to shed some light on why larger learning rates can lead to better generalization in some practical scenarios.

## 1 INTRODUCTION

In the last few years, deep learning has succeeded in establishing state-of-the-art performances in a wide variety of tasks in fields like computer vision, natural language processing and bioinformatics (LeCun et al., 2015). Understanding when and how these networks generalize better is important to keep improving their performance. Many works starting mainly from Neyshabur et al. (2015), Zhang et al. (2017) and Keskar et al. (2017) hint a rich interplay between regularization and the optimization process of learning the weights of the network. The idea is that a form of inductive bias can be realized implicitly by the optimization algorithm. The most popular algorithm to train neural networks is stochastic gradient descent (SGD). It is therefore of great interest to study the generalization properties of this algorithm. An approach that is particularly well suited to investigate learning algorithms directly is the framework of stability (Bousquet & Elisseeff, 2002), (Elisseeff et al., 2005). It is argued in Nagarajan & Kolter (2019) that generalization bounds based on uniform convergence might be condemned to be essentially vacuous for deep networks. Stability bounds offer a possible alternative by trying to bound directly the generalization error of the output of the algorithm. The seminal work of Hardt et al. (2016) exploits this framework to study SGD for both the convex and non-convex cases. The main intuitive idea is to look at how much changing one example in the training set can generate a different trajectory when running SGD. If the two trajectories must remain close to each other then the algorithm has better stability.

This raises the question of how to best measure the distance between two classifiers. Our work investigates a measure of distance respecting invariances in homogeneous neural networks (and linear classifiers) instead of the usual euclidean distance. The measure of distance we consider is directly related to analyzing stability with respect to the normalized loss function instead of the standard loss function used for training. In the convex case, we prove an upper bound on uniform stability with respect to the normalized loss function, which can then be used to prove a high probability bound on the test error of the output of SGD. In the non-convex case, we propose an analysis directly targeted toward homogeneous neural networks. We prove an upper bound on the on-average stability with respect to the normalized loss function, which can then be used to give a generalization bound on the test error. One nice advantage coming with our approach is that we do not need to assume that the loss function is bounded. Indeed, even if the loss function used for training is unbounded, the normalized loss is necessarily bounded.

Our main results for neural networks involve a data-dependent quantity that we estimate during training in our numerical experiments. The quantity is the sum over each layer of the ratio between

the norm of the gradient for this layer and the norm of the parameters for the layer. We observe that larger learning rates lead to trajectories in parameter space keeping this quantity smaller during training. There are two ways to get our data-dependent quantity smaller during training. The first is by facilitating convergence (having smaller norms for the gradients). The second is by increasing the weights of the network. If the weights are larger, the same magnitude for an update in weight space results in a smaller change in angle (see Figure 1). In our experiments, larger learning rates are seen to be more favorable in both regards.

Our main contributions are summarized as follows:

1) The first analysis of stability for SGD directly exploiting invariances in homogeneous neural networks (smooth and non-smooth cases).

2) Empirical observations suggesting that our data-dependent quantity is interesting in understanding how larger learning rates can improve generalization.

3) An analysis of stability for the convex case naturally incorporating the norm of the initial point.

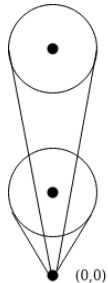

Figure 1: For the same magnitude of step taken (same ball radius), a larger norm of parameters leads to a smaller change in angle.

## 2 RELATED WORK

Normalized loss functions have been considered before (Poggio et al., 2019), (Liao et al., 2018). In Liao et al. (2018), test error is seen to be well correlated with the normalized loss. This observation is one motivation for our study. We might expect generalization bounds on the test error to be better by using the normalized surrogate loss in the analysis. (Poggio et al., 2019) writes down a generalization bound based on Rademacher complexity but motivated by the possible limitations of uniform convergence for deep learning (Nagarajan & Kolter, 2019) we take the stability approach instead.

Generalization of SGD has been investigated before in a large body of literature. Soudry et al. (2018) showed that gradient descent converges to the max-margin solution for logistic regression and Lyu & Li (2019) provides an extension to deep non-linear homogeneous networks. Nacson et al. (2019) gives similar results for stochastic gradient descent. From the point of view of stability, starting from Hardt et al. (2016) without being exhaustive, a few representative examples are Bassily et al. (2020), Yuan et al. (2019), Kuzborskij & Lampert (2018),Liu et al. (2017), London (2017).

Since the work of Zhang et al. (2017) showing that currently used deep neural networks are so overparameterized that they can easily fit random labels, taking properties of the data distribution into account seems necessary to understand generalization of deep networks. In the context of stability, this means moving from uniform stability to on-average stability. This is the main concern of the work of Kuzborskij & Lampert (2018). They develop data-dependent stability bounds for SGD by extending over the work of Hardt et al. (2016). Their results have a dependence on the risk of the initialization point and the curvature of the initialization. They have to assume a bound on the noise of the stochastic gradient. We do not make this assumption in our work. Furthermore, we maintain in our bounds for neural networks the properties after the "burn-in" period and therefore closer to

the final output since we are interested in the effect of the learning rate on the trajectory. This is motivated by the empirical work of Jastrzebski et al. (2020) arguing that in the early phase of training, the learning rate and batch size determine the properties of the trajectory after a "break-even point". Another work interested in on-average stability is Zhou et al. (2021). Differently from our work, their approach makes the extra assumptions that the variance of the stochastic gradients is bounded and also that the loss is bounded. Furthermore, our analysis directly exploits the structure of neural networks and the properties following from using homogeneous non-linearities.

It has been observed in the early work of Keskar et al. (2017) that training with larger batch sizes can lead to a deterioration in test accuracy. The simplest strategy to reduce (at least partially) the gap with small batch training is to increase the learning rate (He et al., 2019), (Smith & Le, 2018), (Hoffer et al., 2017), (Goyal et al., 2017). We choose this scenario to investigate empirically the relevance of our stability bound for SGD on neural networks. Note that the results in Hardt et al. (2016) are more favorable to smaller learning rates. It seems therefore important in order to get theory closer to practice to understand better in what sense larger learning rates can improve stability.

## 3 PRELIMINARIES

Let $l(w, z)$ be a non-negative loss function. Furthermore, let $A$ be a randomized algorithm and denote by $A(S)$ the output of $A$ when trained on training set $S = \{z_1, \cdots, z_n\} \sim \mathcal{D}^n$. The true risk for a classifier $w$ is given as

$$L_{\mathcal{D}}(w) := \mathbb{E}_{z \sim \mathcal{D}} l(w, z)$$

and the empirical risk is given by

$$L_S(w) := \frac{1}{n} \sum_{i=1}^{n} l(w, z_i).$$

When considering the $0 - 1$ loss of a classifier $w$, we will write $L_{\mathcal{D}}^{0-1}(w)$. Furthermore, we will add a superscript $\alpha$ when the normalized losses $l^{\alpha}$ are under consideration (these will be defined more clearly in the subsequent sections respectively for the convex case and the non-convex case). Our main interest is to ensure small test error and so we want to bound $L_{\mathcal{D}}^{0-1}(w)$. The usual approach is to minimize a surrogate loss upper bounding the $0 - 1$ loss. In this paper, we consider stochastic gradient descent with different batch sizes to minimize the empirical surrogate loss. The update rule of this algorithm for learning rates $\lambda_t$ and a subset $B_t \subset S$ of size $B$ is given by

$$w_{t+1} = w_t - \lambda_t \frac{1}{B} \sum_{z_j \in B_t} \nabla l(w_t, z_j). \tag{1}$$

We assume sampling uniformly with replacement in order to form each batch of training examples. In order to investigate generalization of this algorithm, we consider the framework of stability (Bousquet & Elisseeff, 2002).

We now give the definitions for uniform stability and on-average stability (random pointwise hypothesis stability in Elisseeff et al. (2005)) for randomized algorithms (see also Hardt et al. (2016) and Kuzborskij & Lampert (2018)). The definitions can be formulated with respect to any loss function but since we will study stability with respect to the $l^{\alpha}$ losses, we write the definitions in the context of this special case.

**Definition 1** *The algorithm $A$ is said to be $\epsilon_{uni}^{\alpha}$-uniformly stable if for all $i \in \{1, \ldots, n\}$*

$$\sup_{S, z_i', z} \mathbb{E}\left[|l^{\alpha}(A(S), z) - l^{\alpha}(A(S^{(i)}), z)|\right] \le \epsilon_{uni}^{\alpha}. \tag{2}$$

*Here, the expectation is taken over the randomness of $A$. The notation $S^{(i)}$ means that we replace the $i^{th}$ example of $S$ with $z_i'$.*

**Definition 2** *The algorithm $A$ is said to be $\epsilon_{av}^{\alpha}$-on-average stable if for all $i \in \{1, \ldots, n\}$*

$$\mathbb{E}\left[|l^{\alpha}(A(S), z) - l^{\alpha}(A(S^{(i)}), z)|\right] \le \epsilon_{av}^{\alpha}. \tag{3}$$

Here, the expectation is taken over $S \sim \mathcal{D}^n$, $z \sim \mathcal{D}$ and the randomness of A. The notation $S^{(i)}$ means that we replace the $i^{th}$ example of $S$ with $z$.

Throughout the paper, $||\cdot||$ will denote the euclidean norm for vectors and the Frobenius norm for matrices. The proofs are given in Appendix A for the convex case and in Appendix B for the non-convex case.

## 4 CONVEX CASE: A FIRST STEP TOWARD THE NON-CONVEX CASE

Since the convex case is easier to handle, it can be seen as a good preparation for the non-convex case. Consider a linear classifier parameterized by either a vector of weights (binary case) or a matrix of weights (multi-class case) that we denote by $w$ in both cases. The normalized losses are defined by

$$l^\alpha(w, z) := l(\alpha \frac{w}{||w||}, z), \tag{4}$$

for $\alpha > 0$.

In order to state the main result of this section, we need two common assumptions: $L$-Lipschitzness of $l$ as a function of $w$ and $\beta$-smoothness.

**Definition 3** *The function $l(w, z)$ is $L-$Lipschitz for all $z$ in the domain (with respect to $w$) if for all $w, w', z$,*

$$|l(w, z) - l(w', z)| \leq L||w - w'||. \tag{5}$$

**Definition 4** *The function $l(w, z)$ is $\beta-$smooth if for all $w, w', z$,*

$$||\nabla l(w, z) - \nabla l(w', z)|| \leq \beta||w - w'||. \tag{6}$$

We are now ready to state the main result of this section.

**Theorem 1** *Assume that $l(w, z)$ is convex, $\beta-$smooth and $L-$Lipschitz for all $z$. Furthermore, assume that the initial point $w_0$ satisfies $||w_0|| \geq K$ for some $K$ such that $\hat{K} = K - L\sum_{i=0}^{T-1} \lambda_i > 0$ for a sequence of learning rates $\lambda_i \leq 2/\beta$. SGD is then run with batch size $B$ on loss function $l(w, z)$ for $T$ steps with the learning rates $\lambda_t$ starting from $w_0$. Denote by $\epsilon_{uni}^\alpha$ the uniform stability of this algorithm with respect to $l^\alpha$. Then,*

$$\epsilon_{uni}^\alpha \leq \alpha \frac{2L^2 B}{n\hat{K}} \sum_{i=0}^{T-1} \lambda_i. \tag{7}$$

What is the main difference between our bound and the bound in Hardt et al. (2016) (see theorem 7 in Appendix A) ? Our bound takes into account the norm of the initialization. The meaning of the bound is that it is not enough to use small learning rates and a small number of epochs to guarantee good stability (with respect to the normalized loss). We also need to take into account the norm of the parameters (here the norm of the initialization) to make sure that the "effective" learning rates are small. Note that all classifiers are contained in any ball around the origin even if the radius of the ball is arbitrarily small. Therefore, all control over stability is lost very close to the origin where even a small step (in Euclidean distance) can lead to a drastic change in the classifier. The norm of the initialization must therefore be large enough to ensure that the trajectory cannot get too close to the origin (in worst case, since uniform stability is considered). An alternative if the conditions of the theorem are too strong in some practical scenarios is to use on-average stability ($l = 1$ layer in the results of section 5). As a side note, we also incorporated the batch size into the bound which is not present in Hardt et al. (2016) (only $B = 1$ is considered).

From this result, it is now possible to obtain a high probability bound for the test error. The bound is over draws of training sets $S$ but not over the randomness of A. [1] So, we actually have the expected

---

[1]It is possible to obtain a bound holding over the randomness of A by exploiting the framework of Elisseeff et al. (2005). However, the term involving $\rho$ in their theorem 15 does not converge to 0 when the size of the training set grows to infinity.

test error over the randomness of $A$ in the bound. This is reminiscent of PAC-Bayes bounds where here the posterior distribution would be induced from the randomness of the algorithm $A$.

**Theorem 2** *Fix $\alpha > 0$. Let $M_\alpha := \sup\{l(w, z) \text{ s.t. } ||w|| \leq \alpha, ||x|| \leq R\}$. Then, for any $n > 1$ and $\delta \in (0, 1)$, the following hold with probability greater or equal to $1 - \delta$ over draws of training sets $S$:*

$$\mathbb{E}_A L_{\mathcal{D}}^{0-1}(A(S)) \leq \mathbb{E}_A L_S^\alpha(A(S)) + \epsilon_{uni}^\alpha + (2n\epsilon_{uni}^\alpha + M_\alpha)\sqrt{\frac{\ln(1/\delta)}{2n}}. \tag{8}$$

**Proof:** The proof is an application of McDiarmid's concentration bound. Note that we do not need the training loss to be bounded since we consider the normalized loss which is bounded. The proof follows the same line as theorem 12 in Bousquet & Elisseeff (2002) and we do not replicate it here. Note that we need to use that uniform stability implies generalization in expectation which is proven for example in theorem 2.2 from Hardt et al. (2016).

Furthermore, a bound holding uniformly over all $\alpha$'s can be obtained using standard techniques.

**Theorem 3** *Let $C > 0$. Assume that $l^\alpha(w, z)$ is a convex function of $\alpha$ for all $w, z$ and that $\epsilon_{uni}^\alpha$ is a non-decreasing function of $\alpha$. Then, for any $n > 1$ and $\delta \in (0, 1)$, the following hold with probability greater or equal to $1 - \delta$ over draws of training sets $S$:*

$$\mathbb{E}_A L_{\mathcal{D}}^{0-1}(A(S)) \leq \inf_{\alpha \in (0, C]} \left\{ \mathbb{E}_A \max\left(L_S^{\alpha/2}(A(S)), L_S^\alpha(A(S))\right) + \epsilon_{uni}^\alpha + (2n\epsilon_{uni}^\alpha + \right.$$
$$\left. M_\alpha)\sqrt{\frac{2\ln(\sqrt{2}(2 + \log_2 C - \log_2 \alpha)) + \ln(1/\delta)}{2n}} \right\}.$$

In the next section, we investigate the non-convex case. We exploit on-average stability to obtain a data-dependent quantity in the bound. Note that it is also argued in Kuzborskij & Lampert (2018) that the worst case analysis of uniform stability might not be appropriate for deep learning.

## 5 NON-CONVEX CASE

We consider homogeneous neural networks in the setup of multiclass classification. Write $f(x) = W_l(\sigma(\cdots W_2(\sigma(W_1 x))))$, where $x$ is an input to the neural network, $W_i$ denotes the weight matrix at layer $i$ and $\sigma$ denotes a homogeneous non-linearity ($\sigma(cx) = c^k \sigma(x)$ for any constant $c > 0$). Examples for the non-linearity are the ReLU function ($k = 1$), the quadratic function ($k = 2$) and the identity function ($k = 1$, leading to deep linear networks). Consider a non-negative loss function $l(s, y)$ that receives a score vector $s = f(x)$ and a label $y$ as inputs. We require the loss function to be $L$-Lipschitz for all $y$ as a function of $s$. That is, for all $s, s', y$,

$$|l(s, y) - l(s', y)| \leq L||s - s'||. \tag{9}$$

For example, we can use the cross-entropy loss (softmax function with negative log likelihood). In this case, it is simple to show by bounding the norm of the gradient of $l(s, y)$ with respect to $s$ that we can use $L = \sqrt{2}$. Note that this is slightly different from the Lipschitz assumption of the previous section (given with respect to the weights $w$).

In order to control the behaviour of the non-linearity, we assume that for any $c > 0$, there exist constants $B_c$ and $L_c$ such that for any $x, y \in \mathbb{R}^d$ with $||x|| \leq c$ and $||y|| \leq c$ we have

$$||\sigma(x)|| \leq B_c||x||, \tag{10}$$
$$||\sigma(x) - \sigma(y)|| \leq L_c||x - y||. \tag{11}$$

Note that the non-linearity $\sigma$ is being applied component-wise when the input is a vector as above. It is easy to verify that, for the ReLU function, we have $B_c = 1$ and $L_c = 1$ for all $c$. Furthermore, for the quadratic function $x^2$, we have $B_c = c$ and $L_c = 2c$. The following lemma will be the starting point for our analysis.

**Lemma 1** *Assume that $||x|| \leq R$. Let $\alpha_1, \cdots \alpha_l$ be positive real numbers and, for $1 \leq j \leq l$, denote $\tilde{W}_j := \frac{W_j}{||W_j||}$ and $\tilde{W}'_j := \frac{W'_j}{||W'_j||}$. Write $s_j = \alpha_j \tilde{W}_j(\sigma(\cdots \alpha_2 \tilde{W}_2(\sigma(\alpha_1 \tilde{W}_1 x))))$ and $s'_j = \alpha_j \tilde{W}'_j(\sigma(\cdots \alpha_2 \tilde{W}'_2(\sigma(\alpha_1 \tilde{W}'_1 x))))$. Also, let $c_j$ be an upper bound on the norm of layer $j$ (this will be a constant depending on $\alpha_1, \cdots \alpha_j$ and $R$). Then, we have*

$$||s_l - s'_l|| \leq R\left(\prod_{j=1}^{l} \alpha_j\right) \sum_{i=1}^{l} \tau_i ||\tilde{W}_i - \tilde{W}'_i||, \tag{12}$$

*where $\tau_i = \prod_{j=1, j \neq i}^{l} \begin{pmatrix} B_{c_j} & if & j < i \\ L_{c_{j-1}} & if & j > i \end{pmatrix}$.*

The previous lemma motivates a measure of "distance" between neural networks.

**Definition 5** *For neural networks $f$ and $g$, where the weight matrices of $f$ are given by $W_1 \cdots W_l$ and the weight matrices of $g$ are given by $W'_1 \cdots W'_l$, define*

$$d(f, g) := \sum_{i=1}^{l} \tau_i ||\frac{W_i}{||W_i||} - \frac{W'_i}{||W'_i||}||. \tag{13}$$

Note that this distance function is invariant to rescaling the weights of any layer. This is a desirable property since in a homogeneous network such a reparametrization leaves the class predicted by the classifier unchanged for any input to the network.

Let $\alpha_1, \cdots \alpha_l$ be positive real numbers. We define the $l^{\alpha_1, \cdots \alpha_l}(f, z)$ losses to be equal to

$$l(\alpha_l \frac{W_l}{||W_l||}(\sigma(\cdots \alpha_2 \frac{W_2}{||W_2||}(\sigma(\alpha_1 \frac{W_1}{||W_1||} x)))), y), \tag{14}$$

where $z = (x, y)$ and $f$ is the neural network with weight matrix at layer $i$ given by $W_i$. That is, we project the weight matrices to give the norm $\alpha_i$ to layer $i$ and then we evaluate the loss $l$ on this "normalized" network. For simplicity, we will only consider the case where all the $\alpha_i$'s are equal to say $\alpha$ and we will write $l^{\alpha}(f, z)$. From our definitions and lemma 1, we have that for all $z$ and neural networks $f$ and $g$,

$$|l^{\alpha}(f, z) - l^{\alpha}(g, z)| \leq LR\alpha^l d(f, g). \tag{15}$$

In order to bound stability with respect to $l^{\alpha}$, we will have to ensure that the two trajectories cannot diverge too much in terms of $d(f, g)$.

We will consider two separate cases: the smooth case and the non-smooth case. When the activation function is smooth (for example $x^k$ for $k \geq 1$), we will exploit the concept of layer-wise smoothness defined below.

**Definition 6** *Consider the gradient of the loss function with respect to the parameters $W$ for some training example $z$. The vector containing only the partial derivatives for the weights of layer $j$ will be denoted by $\nabla^{(j)} l(W, z)$. We define $\{\beta_j\}_{j=1}^{l}$-layerwise smoothness as the following property: For all $j$, $z$, $W = (W_1, \cdots, W_l)$ and $W' = (W'_1, \cdots, W'_l)$,*

$$||\nabla^{(j)} l(W, z) - \nabla^{(j)} l(W', z)|| \leq \beta_j ||W_j - W'_j||. \tag{16}$$

*We also let $\beta := \max\{\beta_j\}$. Note that $\beta$ is upper bounding the spectral norm of the bloc diagonal approximation of the Hessian.*

We are now ready to state the main theorem of this section for the smooth case.

**Theorem 4** *Suppose that the loss function $l(s,y)$ is $L$-Lipschitz for all $y$, non-negative and that $l^\alpha(f,z)$ is bounded above by $M_\alpha$. Furthermore, assume $\{\beta_j\}_{j=1}^l$-layerwise smoothness and that $||x|| \leq R$. Finally, let $B$ denote the batch size, $\lambda_t \leq \frac{c}{t}$ the learning rates and $T$ the number of iterations SGD is being run. Then,*

$$\epsilon_{av}^\alpha \leq \inf_{t_0 \in \{1,2,\dots,\frac{n}{B}\}} \left[ \frac{2BLR\alpha^l}{(n-B)\beta} \left( \frac{T-1}{t_0-1} \right)^{c\beta} \sum_{t=t_0}^{T-1} \zeta_t + M_\alpha \frac{Bt_0}{n} \right], \qquad (17)$$

*where $\beta = \max\{\beta_j\}$ and $\zeta_t := \sum_{j=1}^l \tau_j \mathbb{E}_{A,S,z} \left[ C_j(S,z)^{T-t} \frac{||\nabla^{(j)} L_{B_t}(W_t)||}{K_t^{(j)}(S,z)} \right]$,*

*with $K_t^{(j)}(S,z) := \min\{||W_{j,t}||, ||W'_{j,t}||\}$ and $C_j(S,z) := \max_{t_0 \leq t \leq T-1} \frac{K_t^{(j)}(S,z)}{K_{t+1}^{(j)}(S,z)}$.*

To evaluate the bound, we need to find the best $t_0$. There is a tradeoff here between two quantities. A small $t_0$ is better for the term $M_\alpha \frac{Bt_0}{n}$ but is worse for the remaining term. This establishes the best "burn in" period. The amount of "exploration" before $t_0$ does not effect the generalization bound. The amount of exploration measured by $\zeta_t$ (and through the learning rate via the value of $c$) becomes important only after iteration $t_0$. The bound will be better if we can reach a region in parameter space such that the classifier is then effectively not changing much. This is measured through the norm of the gradient but also takes into account the norm of the parameters (via $K_t^{(j)}(S,z)$). When we reach larger norms of parameters, stability (with respect to the normalized loss) is less negatively affected. The intuitive reason is the following: the same magnitude of step results in a smaller change in the classifier if the parameters are larger (see Figure 1). In Hoffer et al. (2017), it is observed that small batch training and larger learning rates (finding solutions generalizing better) are reaching larger norms of parameters (see also our Figure 3). Using standard Euclidean distance in the analysis of stability would lead us to believe that this behaviour is highly undesirable. Our analysis shows that this behaviour can actually be favorable to the on-average stability with respect to the normalized loss. The quantity $\zeta_t$ also involves the terms $C_j(S,z)$ measuring how fast the norm of the parameters is growing from one iteration to the next. The value of $C_j(S,z)$ is better (smaller) if the norm of the parameters grows faster.

The non-smooth case is also of great interest since the ReLU activation function is very common in practice.

**Theorem 5** *Suppose that the loss function $l(s,y)$ is $L$-Lipschitz for all $y$, non-negative and that $l^\alpha(f,z)$ is bounded above by $M_\alpha$. Furthermore, assume $||x|| \leq R$. Finally, let $B$ denote the batch size, $\lambda_t$ the learning rates and $T$ the number of iterations SGD is being run. Then,*

$$\epsilon_{av}^\alpha \leq \inf_{t_0 \in \{1,2,\dots,\frac{n}{B}\}} \left[ 2LR\alpha^l \sum_{t=t_0}^{T-1} \lambda_t \zeta_t + M_\alpha \frac{Bt_0}{n} \right], \qquad (18)$$

*where $\zeta_t$ is defined as in theorem 4.*

Exploiting theorem 12 in Elisseeff et al. (2005), it is then possible to get a probabilistic bound on the test error (holding over the randomness in the training sets and the randomness in the algorithm).

**Theorem 6** *Fix $\alpha > 0$. Then, for any $n > 1$ and $\delta \in (0,1)$, the following hold with probability greater or equal to $1 - \delta$ over draws of training sets $S$ and the randomness of the algorithm $A$:*

$$L_\mathcal{D}^{0-1}(A(S)) \leq L_S^\alpha(A(S)) + \sqrt{\left( \frac{1}{\delta} \right) \frac{2M_\alpha^2 + 12nM_\alpha \epsilon_{av}^\alpha}{n}}. \qquad (19)$$

## 6 EXPERIMENTS

### 6.1 LEARNING RATES AND $\zeta_t$

In this section we conduct some experiments on the datasets CIFAR10 Krizhevsky (2009) and MNIST LeCun & Cortes (2010). We consider the scenario where we try to reduce the performance gap

between small batch and large batch training by increasing the learning rate. We will give some evidence suggesting that the quantity $\zeta_t$ can be of interest to assess generalization in this case.

We use a global learning rate being decayed one time by a factor of $10$ in our experiments. No weight decay or momentum is used to stay closer to our theoretical analysis of SGD. Note that in principle, the learning rate could be as large as we want during the inital burn-in period (before $t_0$) without hurting stability. However, this burn-in period must be inside the first epoch in the theoretical results we presented. Since in practice we train for many epochs, it is not clear if such a small burn-in period is long enough to be significant in current practice. We still think that the quantity $\zeta_t$ is relevant to investigate empirically. We approximate its value on a training set $S$ with the quantity $\hat{\zeta}_t(S) := \sum_{j=1}^{l} \frac{||\nabla^j L_{B_t}(W_t)||}{||W_{j,t}||}$. The quantities $C_j(S, z)$ can be evaluated empirically to be very close to $1$ and so we neglect them in the expression for $\hat{\zeta}_t(S)$. Also note that $\tau_j = 1$ for all $j$ in the case of ReLU networks. Instead of plotting the value for each iteration, we average $\hat{\zeta}_t(S)$ for each epoch. This leads to smoother curves.

We use a 5-layer convolutional Relu network consisting in 2 convolutional layers with maxpooling and then 3 fully connected layers with cross-entropy loss on CIFAR10. We use also the cross-entropy loss on MNIST but the neural network is a 6-layers fully connected network. In both cases, we use batch-normalization to facilitate training. All the results in the figures are obtained when using a batch size of 2048. We started by training with a smaller batch size of 256 and then tried to reduce the gap in performance between large batch and small batch training by increasing the learning rate. For example, on CIFAR10, we obtain a test accuracy of $86.23\%$ when using a batch size of 256 and a learning rate of 0.5. When increasing the batch size to 2048 (and maintaining the learning rate to 0.5), the test accuracy dropped to $85.14\%$. This happened even if the training loss reach approximately the same value in both cases (0.0123 for batch size 256 and 0.0167 for batch size 2048). We then increased the learning rate to 1.0 and then to 1.5 reaching $85.63\%$ in both cases (not completely solving the gap but reducing it). A similar phenomenon happens for MNIST. Here, with batch size 256 we get $98.57\%$ ($lr = 0.05$) of test accuracy and for batch size 2048, we get $97.52\%$ ($lr = 0.05$), $98.00\%$ ($lr = 0.1$) and $98.39\%$ ($lr = 0.5$). We plotted the values of $\hat{\zeta}_t(S)$ during training in Figure 2. We can see that it is better during all training when increasing the learning rate.

To compare with the analysis from Hardt et al. (2016), the quantity $\zeta_t$ would be replaced with a global Lipschitz constant which would not be affected by the actual trajectory of the algorithm. Therefore, in comparison to our bound, the bound in Hardt et al. (2016) would be much more favorable to smaller learning rates. In other words, the worst case analysis of uniform convergence would require much smaller learning rates to be used than our result to guarantee good stability. The quantity $\zeta_t$ can be improved by accelerating convergence because of the numerator (norm of the gradients) but also by increasing the denominator (norm of the parameters). A larger learning rate can help in both these regards (see Figure 3). Note also that considering only the norm of the gradients without the norm of the parameters would lead to a less favorable quantity compared to considering both the norm of the gradients and the norm of the parameters. A standard analysis of stability (without the normalized loss) similar to Kuzborskij & Lampert (2018) would not benefit from the norm of the parameters.

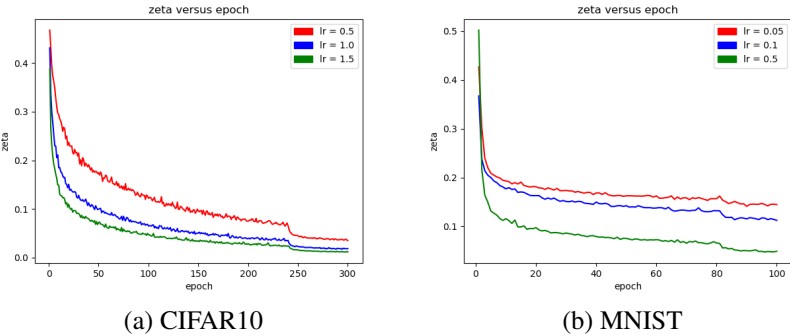

(a) CIFAR10        (b) MNIST

Figure 2: $\hat{\zeta}_t(S)$ when training a convolutional network on CIFAR10 and a fully connected network on MNIST.

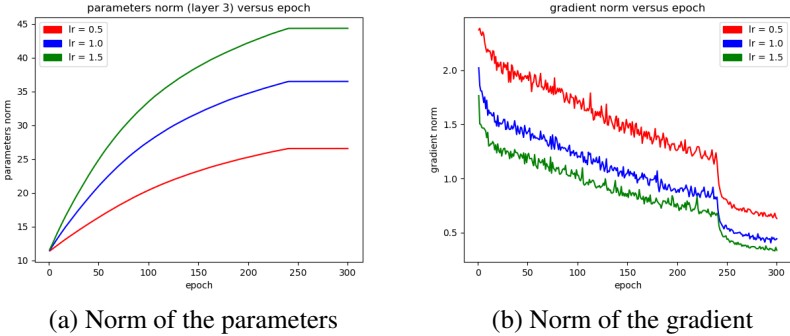

(a) Norm of the parameters        (b) Norm of the gradient

Figure 3: Norm of the parameters (layer 3) and norm of the gradient when training a convolutional network on CIFAR10

## 6.2 GENERALIZATION BOUND AND TEST ERROR

We show in this section the usefulness of considering the normalized loss for bounding the test error. We evaluate the bound in theorem 6 and compare it to an analogous version for the unnormalized loss. For this analogous version, we replace the upper bound $M$ on the loss function by the largest loss achieved during training. Furthermore, the quantity $\epsilon_{av}$ is upper bounded by the Lipschitz constant times the Euclidean distance between the weights of the networks. The Lipschitz constant is replaced by the largest norm of gradients obtained during training. For the normalized loss, we upper bound $\epsilon_{av}^{\alpha}$ by $LR\alpha^l \mathbb{E}d(f, g)$ (see equation 15). We plot the test error, the upper bound for the normalized case with $\alpha = 1.0$ and the upper bound for the unnormalized case in Figure 4. Further experiments (with label noise and a comparison of Adam and SGD) are given in Appendix C.

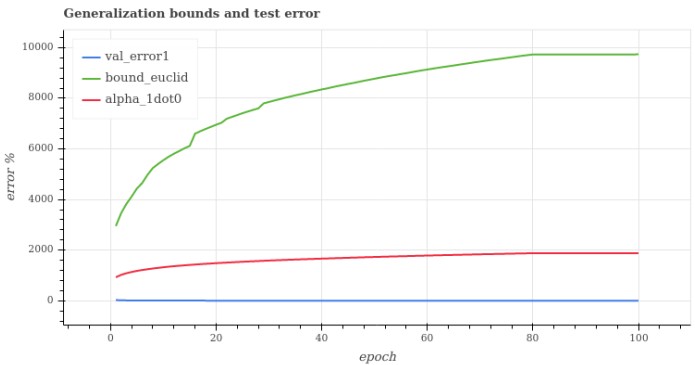

Figure 4: The bound obtained from the Euclidean distance is much worse than the bound obtained from our normalized distance. However, the generalization bound is still vacuous. The network is a 6-layer fully connected network trained on MNIST.

## 7 CONCLUSION

We investigated the stability (uniform and on-average) of $SGD$ with respect to the normalized loss functions. This leads naturally to consider a more meaningful measure of distance between classifiers. Our experimental results show that stability might not be as bad as expected when using larger learning rates in training deep neural networks. We hope that our analysis will be a helpful step in understanding generalization in deep learning. Future work could investigate the on-average stability with respect to $l^{\alpha}$ losses for different optimization algorithms.

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

## A  APPENDIX: PROOFS FOR THE CONVEX CASE

In Hardt et al. (2016), uniform stability with respect to the same loss the algorithm is executed on is considered. This is a natural choice, however if we are interested in the $0 - 1$ loss, different set of parameters $w$, $w'$ can represent equivalent classifiers (that is, predict the same label for any input). This is the case for logistic regression since any rescaling of the parameters yields the same classifier (but they can have different training losses). This is also the case for homogeneous neural networks where we can rescale each layer without affecting the classifier. This is why we consider stability with respect to normalized losses instead. Note that we are still considering SGD executed on the original loss $l$ (we do not change the algorithm $A$). The intuitive idea is to measure stability in terms of angles (more precisely, we consider distances between normalized vectors) instead of standard

euclidean distances (see Figure 1). The proofs in Hardt et al. (2016) consist in bounding $\mathbb{E}||w_t - w'_t||$, where $w_t$ represents the weights at iteration $t$ when training on $S$ and $w'_t$ represents the weights at iteration $t$ when training on the modified training set $S^{(i)}$. We will instead bound $\mathbb{E}||\frac{w_t}{||w_t||} - \frac{w'_t}{||w'_t||}||$ (or $\mathbb{E}[d(f,g)]$ for an appropriate measure of "distance" $d$ between neural networks $f$ and $g$).

**Lemma 2** *Let* $v, w \in \mathbb{R}^n$ *and*
$0 < c \leq \min\{||v||, ||w||\}$. *Then,*

$$||\frac{v}{||v||} - \frac{w}{||w||}|| \leq \frac{||v-w||}{c}.$$

**Proof:** The proof follows from basic linear algebra manipulations. We give it here for completeness since it is important in what follows. We need to show that

$$\langle \frac{v}{||v||} - \frac{w}{||w||}, \frac{v}{||v||} - \frac{w}{||w||} \rangle \leq \frac{\langle v-w, v-w \rangle}{c^2}.$$

After some manipulations, one can see that this is equivalent to show that

$$||v||^2 + ||w||^2 - 2c^2 + 2(c^2 - ||v|| ||w||)\langle \frac{v}{||v||}, \frac{w}{||w||} \rangle \geq 0.$$

From Cauchy-Schwarz inequality, $\langle \frac{v}{||v||}, \frac{w}{||w||} \rangle \leq 1$. Since $c^2 - ||v|| ||w|| \leq 0$, the proof will be completed by showing that

$$||v||^2 + ||w||^2 - 2c^2 + 2(c^2 - ||v|| ||w||) \geq 0.$$

But this is true since

$$||v||^2 + ||w||^2 - 2||v|| ||w|| = (||v|| - ||w||)^2.$$

**Lemma 3** *Assume that the initial point* $w_0$ *satisfies* $||w_0|| \geq K$ *and that SGD is run with batch size* $B$ *and a sequence of learning rates* $\lambda_t$ *on an* $L-$*Lipschitz loss function* $l(w, z)$ *for all* $z$. *Then, for all* $t \geq 1$,

$$||w_t|| \geq K - L\sum_{i=0}^{t-1} \lambda_i.$$

**Proof:**

$$
\begin{aligned}
||w_t|| &= ||w_{t-1} - \lambda_{t-1}\frac{1}{B}\sum_{j=1}^{B} \nabla l(w_{t-1}, z_j)|| \\
&\geq ||w_{t-1}|| - \lambda_{t-1}\frac{1}{B}||\sum_{j=1}^{B} \nabla l(w_{t-1}, z_j)|| \\
&\geq ||w_{t-1}|| - \lambda_{t-1}L \\
&\geq ||w_{t-2}|| - \lambda_{t-2}L - \lambda_{t-1}L \\
&\geq \cdots \\
&\geq ||w_0|| - L\sum_{i=0}^{t-1} \lambda_i \\
&\geq K - L\sum_{i=0}^{t-1} \lambda_i.
\end{aligned}
$$

For ease of comparison, we give the statement of Theorem 3.8 in Hardt et al. (2016).

**Theorem 7** *(Theorem* 3.8 *in Hardt et al. (2016)) Assume that the loss function* $f(\cdot; z)$ *is* $\beta$-*smooth, convex and* $L-$*Lipschitz for every* $z$. *Suppose that we run SGD with step sizes* $\alpha_t \leq 2/\beta$ *for* $T$ *steps. Then, SGD satisfies uniform stability with*

$$\epsilon_{uni} \leq \frac{2L^2}{n} \sum_{t=1}^{T} \alpha_t.$$

We are now ready to prove Theorem 1.

**Proof of Theorem** 1**:** The proof is similar to Hardt et al. (2016). Let $w_t$ denotes the output of $A$ after $t$ steps on training set $S$ and $w'_t$ be the output of $A$ after $t$ steps on training set $S^{(i)}$ for some $i \in \{1, \cdots n\}$. From convexity, the update rule is $1-$expansive (see lemma 3.7 from Hardt et al. (2016)). This property can be used when the example $i$ is not being picked at some iteration. Otherwise, the triangular inequality is used. Since the probability of picking the example $i$ in a mini-batch of size $B$ is smaller than $\frac{B}{n}$ (sampling with replacement) and exploiting Lemma 2 and Lemma 3, we get

$$
\begin{aligned}
\mathbb{E}\left[||\frac{w_{t+1}}{||w_{t+1}||} - \frac{w'_{t+1}}{||w'_{t+1}||}||\right] &\leq \frac{1}{\hat{K}}\mathbb{E}\left[||w_{t+1} - w'_{t+1}||\right] \\
&\leq \frac{1}{\hat{K}}\left[\frac{B}{n}\left(\mathbb{E}||w_t - w'_t|| + 2L\lambda_t\right) + \left(1 - \frac{B}{n}\right)\mathbb{E}||w_t - w'_t||\right] \\
&= \frac{1}{\hat{K}}\left(\mathbb{E}||w_t - w'_t|| + \frac{2BL\lambda_t}{n}\right).
\end{aligned}
$$

Note that this is true since $\mathbb{E}||w_t - w'_t|| \leq \mathbb{E}||w_t - w'_t|| + 2L\lambda_t$. Solving the recursion for $\mathbb{E}||w_t - w'_t||$, we have

$$\mathbb{E}||w_t - w'_t|| \leq \frac{2BL}{n} \sum_{i=0}^{t-1} \lambda_i.$$

Therefore,

$$\mathbb{E}\left[||\frac{w_{t+1}}{||w_{t+1}||} - \frac{w'_{t+1}}{||w'_{t+1}||}||\right] \leq \frac{2BL}{n\hat{K}} \sum_{i=0}^{t} \lambda_i$$

The result then follows from the inequality

$$|l(\alpha\frac{w}{||w||}, z) - l(\alpha\frac{w'}{||w'||}, z)| \leq L\alpha||\frac{w}{||w||} - \frac{w'}{||w'||}||.$$

We finally prove Theorem 3.

**Proof of Theorem** 3**:** To simplify the text, write $\epsilon(\alpha, \delta) := \mathbb{E}_A L_S^\alpha(A(S)) + \epsilon_{stab}^\alpha + (2n\epsilon_{stab}^\alpha + M_\alpha)\sqrt{\frac{\ln(1/\delta)}{2n}}$. For $i \geq 1$, let $\alpha_i = 2^{(1-i)}C$ and $\delta_i = \frac{\delta}{2i^2}$. For any fixed $i$, we have

$$P_S\{\mathbb{E}_A L_{\mathcal{D}}^{0-1}(A(S)) > \epsilon(\alpha_i, \delta_i)\} < \delta_i.$$

Therefore,

$$
\begin{aligned}
&P_S\{\forall i, \ \mathbb{E}_A L_{\mathcal{D}}^{0-1}(A(S)) \leq \epsilon(\alpha_i, \delta_i)\} \\
=\ & 1 - P_S\{\exists i, \ \mathbb{E}_A L_{\mathcal{D}}^{0-1}(A(S)) > \epsilon(\alpha_i, \delta_i)\} \\
\geq\ & 1 - \sum_{i=1}^{\infty} P_S\{\mathbb{E}_A L_{\mathcal{D}}^{0-1}(A(S)) > \epsilon(\alpha_i, \delta_i)\} \\
\geq\ & 1 - \sum_{i=1}^{\infty} \delta_i \geq 1 - \delta.
\end{aligned}
$$

The last inequality follows from

$$\sum_{i=1}^{\infty} \delta_i = \frac{\delta}{2} \sum_{i=1}^{\infty} \frac{1}{i^2} = \frac{\delta}{2}\frac{\pi^2}{6} \leq \delta.$$

We want to show that the set

$$\{S : \; \forall i, \; \mathbb{E}_A L_{\mathcal{D}}^{0-1}(A(S)) \leq \epsilon(\alpha_i, \delta_i)\}$$

is contained in the set

$$\{S : \; \forall \alpha \in (0, C], \; \mathbb{E}_A L_{\mathcal{D}}^{0-1}(A(S)) \leq$$

$$\mathbb{E}_A \max \left( L_S^{\alpha/2}(A(S)), L_S^{\alpha}(A(S)) \right) + \epsilon_{stab}^{\alpha} + (2n\epsilon_{stab}^{\alpha} + M_{\alpha}) \sqrt{\tfrac{2\ln(\sqrt{2}(2+\log_2 C - \log_2 \alpha)) + \ln(1/\delta)}{2n}} \}.$$

Let $S$ be such that $\forall i, \; \mathbb{E}_A L_{\mathcal{D}}^{0-1}(A(S)) \leq \epsilon(\alpha_i, \delta_i)$. Let $\alpha \in (0, C]$. Then, there exists $i$ such that $\alpha_i \leq \alpha \leq 2\alpha_i$. We have

$$\mathbb{E}_A L_{\mathcal{D}}^{0-1}(A(S)) \leq \mathbb{E}_A L_S^{\alpha_i}(A(S)) + \epsilon_{stab}^{\alpha_i} + (2n\epsilon_{stab}^{\alpha_i} + M_{\alpha_i}) \sqrt{\frac{\ln(1/\delta_i)}{2n}}$$

$$\leq \mathbb{E}_A L_S^{\alpha_i}(A(S)) + \epsilon_{stab}^{\alpha} + (2n\epsilon_{stab}^{\alpha} + M_{\alpha}) \sqrt{\frac{\ln(1/\delta_i)}{2n}}$$

$$\leq \mathbb{E}_A L_S^{\alpha_i}(A(S)) + \epsilon_{stab}^{\alpha} + (2n\epsilon_{stab}^{\alpha} + M_{\alpha})$$

$$\sqrt{\frac{2\ln(\sqrt{2}(2 + \log_2 C - \log_2 \alpha)) + \ln(1/\delta)}{2n}}$$

The second inequality is true since both $\epsilon_{stab}^{\alpha}$ and $M_{\alpha}$ are non-decreasing functions of $\alpha$ and $\alpha_i \leq \alpha$. The last inequality is true since $\frac{1}{\delta_i} = \frac{2i^2}{\delta} \leq \frac{2(2+\log_2 C - \log_2 \alpha)^2}{\delta}$. Finally, the proof is concluded by using the convexity of $L_S^{\alpha}(A(S))$ with respect to $\alpha$. Indeed, since $\frac{\alpha}{2} \leq \alpha_i \leq \alpha$, we must have

$$L_S^{\alpha_i}(A(S)) \leq \max \left( L_S^{\alpha/2}(A(S)), L_S^{\alpha}(A(S)) \right).$$

# B  APPENDIX: PROOFS FOR THE NON-CONVEX CASE

**Proof of Lemma** 1**:** The proof is done by induction on the number of layers $l$. Suppose the result is true for $l - 1$ layers. Then we have,

$$||s_l - s_l'|| = \alpha_l ||\tilde{W}_l \sigma(s_{l-1}) - \tilde{W}_l' \sigma(s_{l-1}')||$$

$$= \alpha_l ||\tilde{W}_l \sigma(s_{l-1}) - \tilde{W}_l' \sigma(s_{l-1}) - \tilde{W}_l'(\sigma(s_{l-1}') - \sigma(s_{l-1}))||$$

$$\leq \alpha_l ||\tilde{W}_l \sigma(s_{l-1}) - \tilde{W}_l' \sigma(s_{l-1})|| + \alpha_l ||\tilde{W}_l'(\sigma(s_{l-1}') - \sigma(s_{l-1}))||$$

$$\leq \alpha_l ||\tilde{W}_l - \tilde{W}_l'|| \, ||\sigma(s_{l-1})|| + \alpha_l ||\tilde{W}_l'|| \, ||\sigma(s_{l-1}') - \sigma(s_{l-1})||$$

$$\leq \alpha_l ||\tilde{W}_l - \tilde{W}_l'|| \, ||s_{l-1}|| B_{c_{l-1}} + \alpha_l L_{c_{l-1}} ||s_{l-1}' - s_{l-1}||$$

$$\leq R\alpha_l ||\tilde{W}_l - \tilde{W}_l'|| \prod_{j=1}^{l-1} B_{c_j} \alpha_j + \alpha_l L_{c_{l-1}} ||s_{l-1}' - s_{l-1}||$$

$$\leq R\prod_{j=1}^{l} \alpha_j ||\tilde{W}_l - \tilde{W}_l'|| \prod_{j=1}^{l-1} B_{c_j} + \alpha_l L_{c_{l-1}} R \prod_{j=1}^{l-1} \alpha_j \sum_{i=1}^{l-1} \left[ ||\tilde{W}_i - \tilde{W}_i'|| \prod_{j=1, j\neq i}^{l-1} \left( \begin{array}{cc} B_{c_j} & if \; j < i \\ L_{c_{j-1}} & if \; j > i \end{array} \right) \right]$$

$$= R\prod_{j=1}^{l} \alpha_j \sum_{i=1}^{l} \left[ ||\tilde{W}_i - \tilde{W}_i'|| \prod_{j=1, j\neq i}^{l} \left( \begin{array}{cc} B_{c_j} & if \; j < i \\ L_{c_{j-1}} & if \; j > i \end{array} \right) \right].$$

The proof is finally concluded by observing that for one layer we have, $||s_1' - s_1|| \leq R\alpha_1 ||\tilde{W}_1 - \tilde{W}_1'||$.

**Definition 7** *Let us introduce some notations. Let $\delta_t^{(j)}(S, z) := ||W_{j,t} - W_{j,t}'||$ and $\Delta_t^{(j)}(S, z) := \mathbb{E}_A[\delta_t^{(j)}(S, z) \,|\, \forall k, \; \delta_{t_0}^{(k)}(S, z) = 0]$. Here, $W_{j,t}$ is obtained when training with $S$ for $t$ iterations and $W_{j,t}'$ is obtained when training with $S^{(i)}$ for $t$ iterations. The condition inside the expectation is that after $t_0$ iterations, the two networks are still exactly the same. Since we are interested in*

*distances after normalization, we consider* $\tilde{\delta}_t^{(j)}(S, z) := ||\frac{W_{j,t}}{||W_{j,t}||} - \frac{W'_{j,t}}{||W'_{j,t}||}||$ *and* $\tilde{\Delta}_t^{(j)}(S, z) :=$
$\mathbb{E}_A[\tilde{\delta}_t^{(j)}(S, z) \,|\, \forall k, \, \delta_{t_0}^{(k)}(S, z) = 0]$. *We will further need* $\hat{\delta}_t^{(j)}(S, z) := \frac{\delta_t^{(j)}(S,z)}{K_t^{(j)}(S,z)}$ *and* $\hat{\Delta}_t^{(j)}(S, z) :=$
$\mathbb{E}_A[C_j(S, z)^{T-t}\hat{\delta}_t^{(j)}(S, z) \,|\, \forall k, \, \delta_{t_0}^{(k)}(S, z) = 0]$, *where* $K_t^{(j)}(S, z) := \min\{||W_{j,t}||, ||W'_{j,t}||\}$ *and*
$$C_j(S, z) := \max_{t_0 \le t \le T-1} \frac{K_t^{(j)}(S, z)}{K_{t+1}^{(j)}(S, z)}.$$

Before proving Theorem 4, we establish a lemma. Note that the structure of the proof of the following Lemma and of Theorem 4 is similar to the corresponding results in Hardt et al. (2016) and in Kuzborskij & Lampert (2018).

**Lemma 4** *Suppose that the loss function* $l(s, y)$ *is L-Lipschitz for all* $y$, *non-negative and that* $l^\alpha(f, z)$ *is bounded above by* $M_\alpha$. *Also, assume that* $||x|| \le R$. *Furthermore, let* $B$ *denote the batch size and* $T$ *the number of iterations SGD is being run. Then, for any* $t_0 \in \{0, 1, 2, \ldots, \frac{n}{B}\}$, *the on-average stability satisfies*

$$\epsilon_{av}^\alpha \le LR\alpha^l \sum_{j=1}^{l} \tau_j \mathbb{E}_{S,z}\left[\mathbb{E}_A[\tilde{\delta}_T^{(j)}(S, z) \,|\, \forall k, \, \delta_{t_0}^{(k)}(S, z) = 0]\right] + M_\alpha\left(\frac{Bt_0}{n}\right).$$

**Proof:** Write the quantity $|l^\alpha(f, z) - l^\alpha(g, z)|$ as the sum of $|l^\alpha(f, z) - l^\alpha(g, z)|I\{\forall k, \, \delta_{t_0}^{(k)}(S, z) = 0\}$ and $|l^\alpha(f, z) - l^\alpha(g, z)|I\{\exists k : \, \delta_{t_0}^{(k)}(S, z) \ne 0\}$. We bound the first term by using the fact that

$$|l^\alpha(f, z) - l^\alpha(g, z)| \le LR\alpha^l d(f, g) = LR\alpha^l \sum_{j=1}^{l} \tau_j \tilde{\delta}_T^{(j)}(S, z).$$

For the second term, we use that $l^\alpha(f, z)$ is bounded above by $M_\alpha$ and non-negative to write

$$|l^\alpha(f, z) - l^\alpha(g, z)| \le M_\alpha.$$

The result then follows from the fact that the probability of picking example $i$ in $t_0$ iterations is smaller than $\frac{Bt_0}{n}$.

**Proof of theorem** 4: From Lemma 2, we always have $\tilde{\delta}_t^{(j)}(S, z) \le \hat{\delta}_t^{(j)}(S, z)$. Therefore, from the previous Lemma,

$$\epsilon_{av}^\alpha \le LR\alpha^l \sum_{j=1}^{l} \tau_j \mathbb{E}_{S,z}\left[\hat{\Delta}_T^{(j)}(S, z)\right] + M_\alpha\left(\frac{Bt_0}{n}\right).$$

First note that under our definitions,

$$\begin{aligned}
\hat{\delta}_{t+1}^{(j)}(S, z) &= \frac{\delta_{t+1}^{(j)}(S, z)}{K_{t+1}^{(j)}(S, z)} \\
&\le C_j(S, z)\frac{\delta_{t+1}^{(j)}(S, z)}{K_t^{(j)}(S, z)} \\
&\le \frac{C_j(S, z)}{K_t^{(j)}(S, z)}\left[\delta_t^{(j)}(S, z) + \lambda_t||\nabla^{(j)} L_{B_t}(W_t) - \nabla^{(j)} L_{B'_t}(W'_t)||\right] \\
&= C_j(S, z)\,\hat{\delta}_t^{(j)}(S, z) + \lambda_t C_j(S, z)\frac{||\nabla^{(j)} L_{B_t}(W_t) - \nabla^{(j)} L_{B'_t}(W'_t)||}{K_t^{(j)}(S, z)}.
\end{aligned}$$

Therefore,

$$C_j(S,z)^{T-(t+1)}\hat{\delta}_{t+1}^{(j)}(S,z) \leq C_j(S,z)^{T-t}\hat{\delta}_t^{(j)}(S,z)$$
$$+ \lambda_t C_j(S,z)^{T-t}\frac{||\nabla^{(j)}L_{B_t}(W_t) - \nabla^{(j)}L_{B_t'}(W_t')||}{K_t^{(j)}(S,z)}.$$

Here, $B_t$ denotes the batch of samples at iteration $t$ when training on $S$ and $B_t'$ denotes the batch of samples at iteration $t$ when training on $S^{(i)}$. When $B_t = B_t'$, we will use $\{\beta_j\}_{j=1}^l$-layerwise smoothness to bound $||\nabla^{(j)}L_{B_t}(W_t) - \nabla^{(j)}L_{B_t'}(W_t')||$. Otherwise, we use simply the triangular inequality. Let $p(B,n)$ be the probability of picking the example $i$ in a mini-batch of size $B$ (this is smaller than $\frac{B}{n}$). For $t \geq t_0$, we have

$$\hat{\Delta}_{t+1}^{(j)}(S,z) \leq (1-p(B,n))(1+\beta_j\lambda_t)\hat{\Delta}_t^{(j)}(S,z) + p(B,n)\Big(\hat{\Delta}_t^{(j)}(S,z) +$$
$$\lambda_t \mathbb{E}_A[C_j(S,z)^{T-t}\frac{||\nabla^{(j)}L_{B_t}(W_t)||}{K_t^{(j)}(S,z)} + C_j(S,z)^{T-t}\frac{||\nabla^{(j)}L_{B_t'}(W_t')||}{K_t^{(j)}(S,z)}]\Big).$$

Define $\hat{\Delta}_t^{(j)} := \mathbb{E}_{S,z}\hat{\Delta}_t^{(j)}(S,z)$ and
$\zeta_t^{(j)} := \mathbb{E}_{A,S,z}C_j(S,z)^{T-t}\frac{||\nabla^{(j)}L_{B_t}(W_t)||}{K_t^{(j)}(S,z)}$ for any $t$. Taking the expectation over $S$ and $z$ on both sides of the previous inequality, we get

$$\hat{\Delta}_{t+1}^{(j)} \leq (1-p(B,n))(1+\beta_j\lambda_t)\hat{\Delta}_t^{(j)} + p(B,n)(\hat{\Delta}_t^{(j)} + 2\lambda_t\zeta_t^{(j)}).$$

This is true since
$\mathbb{E}_{A,S,z}C_j(S,z)^{T-t}\frac{||\nabla^{(j)}L_{B_t}(W_t)||}{K_t^{(j)}(S,z)} = \mathbb{E}_{A,S,z}C_j(S,z)^{T-t}\frac{||\nabla^{(j)}L_{B_t'}(W_t')||}{K_t^{(j)}(S,z)}$. Rearranging terms and using $1+x \leq \exp(x)$, we get

$$\hat{\Delta}_{t+1}^{(j)} \leq [1+(1-p(B,n))\beta_j\lambda_t]\hat{\Delta}_t^{(j)} + 2p(B,n)\lambda_t\zeta_t^{(j)}$$
$$\leq \exp((1-p(B,n))\beta_j\lambda_t)\hat{\Delta}_t^{(j)} + 2p(B,n)\lambda_t\zeta_t^{(j)}.$$

Developing the recursion yields

$$\hat{\Delta}_T^{(j)} \leq \sum_{t=t_0}^{T-1} 2p(B,n)\lambda_t\zeta_t^{(j)} \prod_{k=t+1}^{T-1} \exp\left((1-p(B,n))\frac{c\beta_j}{k}\right)$$

$$\leq \frac{2B}{n}\sum_{t=t_0}^{T-1}\lambda_t\zeta_t^{(j)}\exp\left((1-p(B,n))c\beta_j\sum_{k=t+1}^{T-1}\frac{1}{k}\right)$$

$$\leq \frac{2B}{n}\sum_{t=t_0}^{T-1}\lambda_t\zeta_t^{(j)}\exp\left((1-p(B,n))c\beta_j\log(\frac{T-1}{t})\right)$$

$$\leq \frac{2Bc}{n}\sum_{t=t_0}^{T-1}\frac{\zeta_t^{(j)}}{t}\left(\frac{T-1}{t}\right)^{(1-p(B,n))c\beta_j}$$

$$\leq \frac{2Bc}{n}\sum_{t=t_0}^{T-1}\frac{\zeta_t^{(j)}}{t}\left(\frac{T-1}{t}\right)^{(1-p(B,n))c\beta}$$

$$\leq \frac{2Bc}{n}\max_{t_0 \leq t \leq T-1}\{\zeta_t^{(j)}\}(T-1)^{(1-p(B,n))c\beta}\sum_{t=t_0}^{T-1}\left(\frac{1}{t}\right)^{(1-p(B,n))c\beta-1}$$

$$\leq \frac{2Bc}{nc(1-p(B,n))\beta}\max_{t_0 \leq t \leq T-1}\{\zeta_t^{(j)}\}\left(\frac{T-1}{t_0-1}\right)^{(1-p(B,n))c\beta}$$

$$\leq \frac{2B}{(n-B)\beta}\left(\frac{T-1}{t_0-1}\right)^{c\beta}\max_{t_0 \leq t \leq T-1}\{\zeta_t^{(j)}\}.$$

Therefore, $\epsilon_{av}^{\alpha}$ is upper bounded by

$$\inf_{t_0 \in \{1,2,\ldots,\frac{n}{B}\}} \left[ \frac{2BLR\alpha^l}{(n-B)\beta} \left( \frac{T-1}{t_0-1} \right)^{c\beta} \sum_{j=1}^{l} \tau_j \max_{t_0 \leq t \leq T-1} \{\zeta_t^{(j)}\} + M_\alpha(\frac{Bt_0}{n}) \right].$$

To complete the proof, we will use that $\max_{t_0 \leq t \leq T-1}\{\zeta_t^{(j)}\} \leq \sum_{t=t_0}^{T-1} \zeta_t^{(j)}$ and reverse the sum order. With the definition $\zeta_t := \sum_{j=1}^{l} \tau_j \zeta_t^{(j)}$, we then have

$$\epsilon_{av}^{\alpha} \leq \inf_{t_0 \in \{1,2,\ldots,\frac{n}{B}\}} \left[ \frac{2BLR\alpha^l}{(n-B)\beta} \left( \frac{T-1}{t_0-1} \right)^{c\beta} \sum_{t=t_0}^{T-1} \zeta_t + M_\alpha(\frac{Bt_0}{n}) \right].$$

**Proof of Theorem 5:** The beginning of the proof is the same as the proof of Theorem 4. However, in the case where smoothness is not assume (for example, ReLU neural networks), it is not possible to exploit the property of layer-wise smoothness. Instead, only the triangular inequality is used to bound $||\nabla^{(j)} L_{B_t}(W_t) - \nabla^{(j)} L_{B_t'}(W_t')||$. This leads to the inequality

$$\hat{\Delta}_{t+1}^{(j)} \leq \hat{\Delta}_t^{(j)} + 2\lambda_t \zeta_t^{(j)}.$$

Solving the recursion then yields

$$\hat{\Delta}_T^{(j)} \leq 2 \sum_{t=t_0}^{T-1} \lambda_t \zeta_t^{(j)}.$$

As a consequence,

$$\epsilon_{av}^{\alpha} \leq \inf_{t_0 \in \{1,2,\ldots,\frac{n}{B}\}} \left[ 2LR\alpha^l \sum_{t=t_0}^{T-1} \lambda_t \zeta_t + M_\alpha \frac{Bt_0}{n} \right],$$

where $\zeta_t = \sum_{j=1}^{l} \tau_j \zeta_t^{(j)}$, concluding the proof.

## C  APPENDIX: MORE EXPERIMENTS

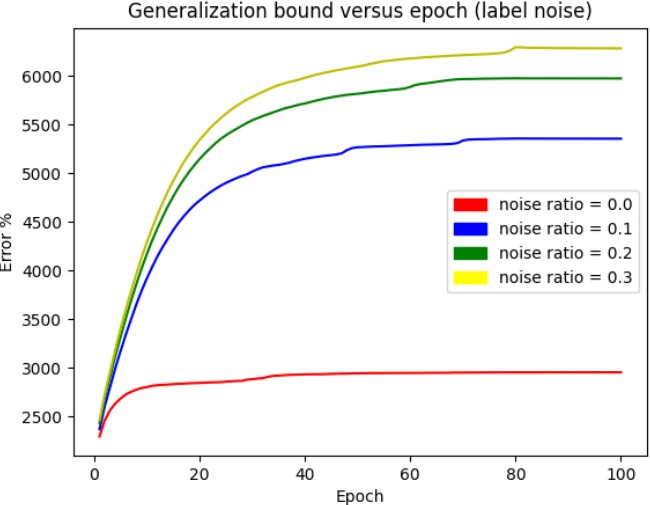

Figure 5: The generalization bound from Theorem 6 ($\alpha = 1.0$) when training with different amounts of label noise. The network is a 6-layer fully connected network trained on MNIST with SGD. The final test accuracies are: $98.56\%$ (label noise ratio $= 0.0$), $96.06\%$ (label noise ratio $= 0.1$), $91.28\%$ (label noise ratio $= 0.2$) and $84.22\%$ (label noise ratio $= 0.3$).

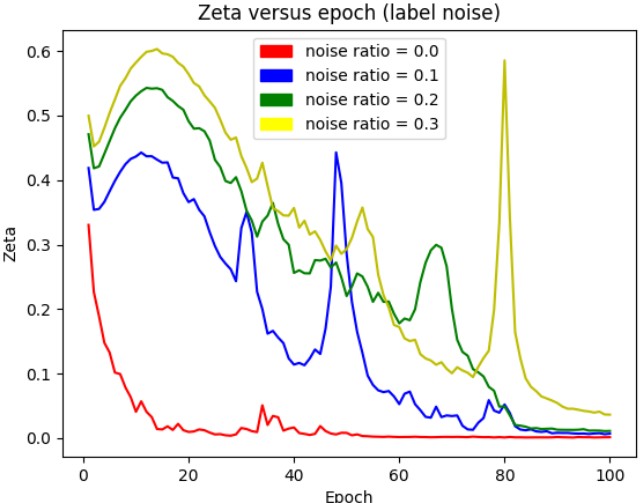

Figure 6: $\hat{\zeta}_t(S)$ when training with different amounts of label noise. The network is a 6-layer fully connected network trained on MNIST with SGD.

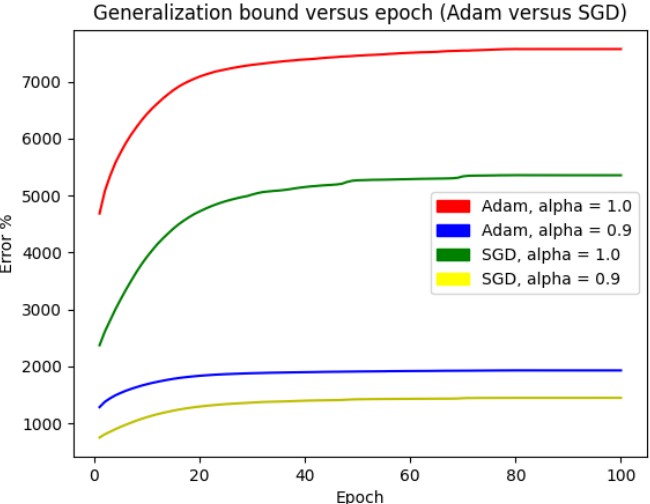

Figure 7: The generalization bound from theorem 6 ($\alpha = 1.0$ and $\alpha = 0.9$) is estimated when training with Adam and SGD. The network is a 6-layer fully connected network trained on MNIST with $10\%$ of label noise. For Adam the test accuracy is $95.38\%$ and for SGD, the test accuracy is $96.06\%$. Those are the best test accuracies that could be obtained after tuning the learning rate for each respective algorithm.

