# OpenReview forum: "Stability analysis of SGD through the normalized loss function"
_ICLR.cc/2022/Conference — ICLR 2022 Submitted_

### Official Review · Reviewer_MY7B · 2021-11-02

**Correctness:** 4
**Technical Novelty And Significance:** 2
**Empirical Novelty And Significance:** 2
**Recommendation:** 5
**Confidence:** 4

**Main Review:**

1- The definition of the stability is for normalized surrogate loss. For instance we might measure the performance with 0-1 loss but train using the cross-entropy loss. I could not find the result in the paper that relate the stability parameter of normalized surrogate loss to 0-1 loss? Note in the Hardt el al paper, we only have one loss function. For instance, in Theorem 2 of  Kuzborskij and Lampert, you can find a such a result.

2- In general, I find it difficult to interpret the results in this paper. Theorem 2 implies that if the initialization point has a very large norm then it helps generalization. Could you please provide more intuition behind this result? Also, please compare your result with Theorem 3 of  Kuzborskij and Lampert, where their bound basically implies that if the distance between the initialization and the optimal point is small, then the sgd is stable for convex case.

3-A technical question is can you provide more details to proof of Theorem 2. Specifically, how do you relate the normalized loss to 0-1 loss?

4-  In the definition of zeta_t, what prevents the denominator to not be close to zero while the numerator be bounded away from zero?

5-  The main quantity for the non-convex setting is the zeta. I could not find any intuition behind this term? Also, how can we compare it with Lipschitz constant?

6- In your experiment you used the batch normalization. However, the theoretical result is obtained for vanilla SGD.

7- In general the figures quality is not good. For instance in Figure 4, the legend and labels are not clear at all.

8- In general, there are lots of "proposal" for the relevant complexity measure for generalization. It would be nice to compare your zeta with the proposals. Please check the Table 1 of "Uniform convergence may be unable to explain generalization in deep learning" by Nagarajan, Kolter to find a list of such measures.

**Summary Of The Paper:**

This paper considers the problem of understanding the generalization of SGD using the stability framework. The well-known result in this line of work is the paper by Hardt'16. In Hardt'16, the stability is measured using the difference between the "actual" weights of two copy of SGD which differ in a single data point. The main observation by the authors in this paper is that in many cases, the loss function is invariant to the scaling of weights. Then, they reformulate the stability analysis using the "normalized loss function" which is defined by  l^alpha(w,z) = loss(alpha*w/||w||,z) where alpha is a constant.

Their main results are the new stability analysis for this new notion for convex and non-convex settings. Specifically, for the convex case the analysis is very similar to the Hardt paper. For the non-convex the authors define a new measure for generalization "zeta" in Theorem 4 which governs the stability.

**Summary Of The Review:**

Interesting motivation and direction however the paper is not well written and there is no discussion for the results. Also, the numerical results section can be improved greatly.

---

> ### Author Response · Authors · 2021-11-19
> **Answers to your questions**
>
> Thank you for your time reviewing our paper. We offer some clarifications below:
>
> 1) and 3. The following is true for homogeneous neural networks:
>
> $L_{\mathcal{D}}^{0-1}(W) = L_{\mathcal{D}}^{0-1}(\tilde{W})\leq L_{\mathcal{D}}(\tilde{W})= L_S(\tilde{W}) + (L_{\mathcal{D}}(\tilde{W})- L_S(\tilde{W}))$,
>
> where $\tilde{W}$ represents the weights after normalization. The first equality follows from positive homogeneity. The inequality follows from the fact that the surrogate loss is upper bounding the $0-1$ loss. Bounding the term $L_{\mathcal{D}}(\tilde{W})- L_S(\tilde{W})$ means in words that we want to control the extent to which we will overfit the normalized loss. This is done with the stability analysis . I think that what you are looking for is exactly Theorem 2.2 in Hardt applied to the special case of the normalized loss (we mention this in the outline for proof of Theorem $2$). It is really the same thing.
>
> 2) In order to prevent overfitting, we want the algorithm to be limited in its ability to explore the space of hypothesis (''effectively'' leading to a smaller hypothesis class). In the context of the normalized loss, the geometric intuition is that exploration is measured with variations in angle instead of euclidean distances. Consider some initial point $w_0$. Theorem $1$ essentially says that if we use smaller learning rates and a larger norm of $w_0$, the possible change in angle will be more restricted. Consider the case where we let the norm of $w_0$ grows to infinity and everything else is fixed (learning rates, T). Since the norm of the gradient is bounded by $L$, the change in angle will be very small after a step taken by SGD if the norm of the parameters is large enough. At infinity, we would remain at $w_0$, that is SGD would return the initial point. No exploration is done and the upper bound in equation $7$ is $0$.
>
> The bound on stability in Kuzborskij and Lampert (on-average stability, our Theorem 1 is for uniform stability) involves the risk of the initial point. We first remark that this risk is not known (it is the true risk and not the empirical risk). The intuition is that if the initial point is already good, we can expect good generalization. A similar intuition can be drawn from our result. Indeed, if the empirical normalized loss of the initial point is already small and we cannot travel too far from it in terms of angles then the algorithm will generalize well.
> We want to remark also that Kuzborskij and Lampert does not offer probabilistic bounds (only bounds on on-average stability) contrary to us.
>
> 4) The numerator will converge to $0$ if SGD converges to any local minimum. The idea is that the quantity $\zeta_t$ is data-dependent and therefore we hope that the particular problem will behave good enough in order to have small values for $\zeta_t$. This is verified empirically on CIFAR10 and MNIST in Figure 2. Indeed, $\zeta_t$ converges to $0$ when $t$ increases in these two cases.
>
> 5) The Lipschitz  constant (with respect to the parameters) is an upper bound on the norm of the gradient. The quantity $\zeta_t$ involves the actual norm of the gradient and this will be better than the worst case Lipschitz constant. This is an advantage of on-average stability. As we discuss below Theorem $4$, the intuition behind $\zeta_t$ is as a measure of exploration of the hypothesis class.
>
> 6) This is true. Training is much harder without it.
>
> 7) We can increase a little bit the size of the figures. Also the legend of Figure $4$ could be more clear. The green curve is the bound using the euclidean distance, the red curve is the bound with our method and $\alpha=1$ and finally the blue curve is the test error.
>
> 8) It is a good suggestion to add more experiments by comparing to other measures of generalization.

---

> > ### Comment · Reviewer_MY7B · 2021-11-29
> > **Thanks**
> >
> > I would like to thank the authors for their response. I still do not have a good understanding of the normalized loss function. I think this paper is indeed interesting as it provides a new framework to analyze generalization based on the normalized loss. However, with the current presentation it is difficult to appreciate the results.
> >
> > Also, for instance, in the revised version of the paper I have not seen new set of experiments regarding my Q8.
> >
> > Best.

---

> > > ### Author Response · Authors · 2021-12-01
> > > **New experiments**
> > >
> > > We are in the process of doing new experiments. In Nagarajan and Kolter, it is argued that the bounds of Neyshabur18 and Bartlett17 do not behave properly when increasing the training set size. Indeed, the bounds grow with the sample size. This comes from the fact that the norm of the parameters of the solution are larger for larger sample sizes. We can expect our method to behave better since larger norms of the parameters are not a problem with our bounds (see also our answer to reviewer hQwz). An experiment on CIFAR10 with different sample sizes shows that our bound indeed decreases when increasing the sample size.
> > >
> > > Let us know what you think is difficult to grasp about the normalized loss function. It might help us improve the clarity of the explanations.
> > >
> > > Best.

---

### Official Review · Reviewer_hQwz · 2021-11-03

**Correctness:** 2
**Technical Novelty And Significance:** 1
**Empirical Novelty And Significance:** 1
**Recommendation:** 3
**Confidence:** 4

**Main Review:**

My main concern about this paper is its motivation for considering/highlighting the normalized loss function. To me, it is a serious concern as all the analysis conducted in this paper is built upon the normalized loss. However, here the normalization, in my opinion, is more of a trick in the analysis. I read through the two papers, Poggio et al., 2019 and Liao et al., 2018, which, according to the authors, introduced normalized loss function. However, it seems that the normalization was used only to simplify the analysis and similar theoretical results could be obtained without normalization. Such comments may also apply to the present paper. In short, I cannot agree with the authors that it is the normalization operation that leads to the results developed in this paper.

In addition, in my opinion, the presentation of the paper needs to be greatly improved. For instance, the authors highlighted the normalized loss throughout, including the title. The notation of normalized loss is also used in Definitions 1 and 2 from Section 3. However, the definition of normalized loss function was only introduced in Section 4. I believe that the normalized loss is not something well-established in the literature. From the two papers mentioned by the authors, I failed to find an explicit definition of the normalized loss function. Therefore, I think the authors really need to define the loss before using the concept again and again. On the other hand, when stating the motivation and the origin of the normalized loss function, the authors only mentioned them in passing at the beginning of Section 2. It seems to me that the authors said a lot about the literature but were trying to avoid answering the question "why this work". I do think that the authors need to take care of this part more seriously.

**Summary Of The Paper:**

This paper conducted a stability analysis of Stochastic Gradient Descent (SGD) for empirical risk minimization induced by the so-called normalized loss function. Here, the normalization is taken with respect to parameters involved in an individual loss; see (4) for the definition of the normalized loss function. The paper should be regarded as a theoretical paper. The main results are stability bounds of SGD for convex and nonconvex ERM schemes.

**Summary Of The Review:**

Based on my comments above, I cannot recommend its acceptance.

---

> ### Author Response · Authors · 2021-11-18
> **Motivation and Presentation**
>
> Thank you for your time. We provide below some discussion about what we think are your two main concerns: Motivation (''why this work'') and presentation.
>
> $1)$ Motivation:
> Consider the following algorithm: phase 1; run SGD as usual, phase 2; multiply all the weights by a large constant at the end of training. This algorithm effectively outputs the same classifier as SGD. The training loss can be assumed to be very close to $0$ for both SGD and this modified version of SGD (under overparametrization). In a generalization bound of the form $L_{\mathcal{D}}^{0-1}(A(S))\leq L_S(A(S))+ Reg$, the determining factor to differentiate the two aforementioned algorithms will then be $Reg$ (since in both cases $L_S(A(S))\approx 0$). Under a standard stability analysis for the surrogate loss (used for training), the term $Reg$ will be much worse for the modified algorithm. Indeed, the proof will be bounding $||w-w'||$ and if $C$ is the constant used to multiply all the weights then the term $Reg$ will be multiplied also by $C$. Choosing $C$ large enough, the bound will become as bad as we want. However, the $0-1$ test loss is the same for both algorithms. Our approach does not suffer from this problem. Indeed,      our analysis leads to the same generalization bound for both SGD and this modified version of SGD. Even though this modified version of SGD would not be used in practice, it still illustrates a problem that would occur if the different directions SGD takes happen to have a high component in the direction of increasing weights. The term $Reg$ would be very large, thus leading to a bad bound.
>
> The following is true for homogeneous neural networks:
> $L_{\mathcal{D}}^{0-1}(W) = L_{\mathcal{D}}^{0-1}(\tilde{W})\leq L_{\mathcal{D}}(\tilde{W})= L_S(\tilde{W}) + (L_{\mathcal{D}}(\tilde{W})- L_S(\tilde{W}))$,
>
> where $\tilde{W}$ represents the weights after normalization.
> The use of the normalized loss function can be seen as the way our analysis exploits the properties of homogeneous neural networks. The property of positive homogeneity is crucial since otherwise the first equality above is not true. Also, the normalization is layer-wise and this reflects how our analysis is done in a layer-wise fashion.
> We also want to emphasize that our analysis leads to quantitatively different results from a standard analysis based on euclidean distance (see Figure 6). In essence, our paper uses the framework of stability to upper bound the quantity $L_{\mathcal{D}}(\tilde{W})- L_S(\tilde{W})$. This leads to a measure of distance between neural networks that is invariant to rescaling the weights of any layer. Because of the normalization, the problem raised in the first paragraph is circumvented.
>
> $2)$ Presentation: We mention in the preliminaries (section $3$) that the normalized loss functions will be defined in the subsequent sections. However, we could move the definitions in section $3$ already to improve clarity. Also, we can add a short description even earlier (in the introduction).
>
> If you want more clarifications on some other points, we will be happy to provide them. Also, if you have explicit examples of incorrect or not well-supported claims, we will be happy to address them.

---

### Official Review · Reviewer_WveE · 2021-11-05

**Correctness:** 4
**Technical Novelty And Significance:** 3
**Empirical Novelty And Significance:** 3
**Recommendation:** 8
**Confidence:** 3

**Main Review:**

The paper is generally very well written. The idea of using normalized losses and angle-wise stability to analyze SGD is interesting and novel. One minor concern I have is the motivation/justification behind using angle-wise stability and overall applicability in neural network learning. Authors claim that in the framework of on-average stability and normalized losses, larger norms of parameters are beneficial to stability. However, it is known that this is probably not generally the case. Even though authors show some empirical evidence in Fig 3, this argument can be better motivated.

**Summary Of The Paper:**

This paper provides new generalization bounds for SGD using the stability analysis framework for both convex and non-convex cases using a normalized loss function, stability as measured in the form of anglewise stability as supposed to standard euclidean distance.

**Summary Of The Review:**

I feel the theoretical results are strong and the authors provide sufficiant empirical results to validate their theoretical findings.

---

> ### Author Response · Authors · 2021-11-18
> **More explanations**
>
> Thank you for time reading our paper. We provide a few more explanations below in the hope of bringing further clarifications.
>  The main intuition for the stability framework is that less exploration of the hypothesis space by the optimization algorithm will lead to less overfitting. Since unconstrained SGD can find solutions with large norms of the parameters, the standard analysis (based on euclidean distance) would lead to bad bounds on the generalization error. However, the invariance properties of homogeneous neural networks imply that increasing the weights do not really constitutes a form of exploration of the hypothesis class. This leads us to consider a form of angle-wise stability instead of Euclidean stability. The reason why larger norms of parameters can be beneficial to angle-wise stability is  that, in such a region, the angle between successive iterations cannot change a lot (for the same size of step taken). If the empirical normalized loss is the same for two different neural networks, we might expect that the one found with the smallest amount of exploration will generalize better. The data-dependent quantity involved in ''measuring'' this amount of exploration in our results is $\zeta_t$.

---

### Decision · Program_Chairs · 2022-01-20

**Decision:**

Reject

**Comment:**

The paper focuses on providing generalization bounds for SGD for functions that are invariant under scaling. The paper's analysis is based on the stability framework but instead focuses on a metric that is based on the anglular distance as compared to the euclidean distance.

Overall the reviewers found the paper to be interesting and the results to be useful. However the reviewers found the paper to be significantly lacking in terms of its presentation. In particular a clear exposition of the central object of the paper, i.e. normalized loss function was missing as well as clear comparisons between the presented results and existing results. I recommend the authors to motivate their results better and contrast their presented results with existing results to fully highlight the impact of their presented result. Hopefully the suggestions made by the reviewers in terms of presentation will be helpful to the authors towards improving the paper.